



# Identifying the sources of uncertainty in climate model simulations of solar radiation modification with the G6sulfur and G6solar Geoengineering Model Intercomparison Project (GeoMIP) simulations

Daniele Visioni[1], Douglas G. MacMartin[1], Ben Kravitz[2,3], Olivier Boucher[4], Andy Jones[5],
Thibaut Lurton[4], Michou Martine[6], Michael J. Mills[7], Pierre Nabat[6], Ulrike Niemeier[8],
Roland Séférian[6], and Simone Tilmes[7]

[1]Sibley School for Mechanical and Aerospace Engineering, Cornell University, Ithaca, NY, USA
[2]Department of Earth and Atmospheric Science, Indiana University, Bloomington, IN, USA
[3]Atmospheric Sciences and Global Change Division, Pacific Northwest National Laboratory, Richland, WA, USA
[4]Institut Pierre-Simon Laplace, Sorbonne Université/CNRS, Paris, France
[5]Met Office Hadley Centre, Exeter, EX1 3PB, UK
[6]CNRM, Université de Toulouse, Météo-France, CNRS, Toulouse, France
[7]Atmospheric Chemistry, Observations, and Modeling Laboratory, National Center for Atmospheric Research, Boulder, CO, USA
[8]Max Planck Institute for Meteorology, Hamburg, Germany

**Correspondence:** Daniele Visioni (dv224@cornell.edu)

**Abstract.** We present here results from the Geoengineering Model Intercomparison Project (GeoMIP) simulations for the experiment G6sulfur and G6solar for six Earth System Models participating in the Climate Model Intercomparison Project (CMIP) Phase 6. The aim of the experiments is to reduce the warming from that resulting from a high-tier emission scenario (Shared Socioeconomic Pathways SSP5-8.5) to that resulting from a medium-tier emission scenario (SSP2-4.5). These simu-

lations aim to analyze the response of climate models to a reduction in incoming surface radiation as a means to reduce global surface temperatures, and they do so either by simulating a stratospheric sulfate aerosol layer or, in a more idealized way, through a uniform reduction in the solar constant in the model. We find that, by the end of the century, there is a considerable inter-model spread in the needed injection of sulfate ($29 \pm 9$ Tg-$SO_2$/yr between 2081 and 2100), in how the aerosol cloud is distributed latitudinally, and in how stratospheric temperatures are influenced by the produced aerosol layer. Even in the

simpler G6solar experiment, there is a spread in the needed solar dimming to achieve the same global temperature target ($1.91 \pm 0.44$ %). The analyzed models already show significant differences in the response to the increasing $CO_2$ concentrations for global mean temperatures and global mean precipitation ($2.05$K $\pm 0.42$K and $2.28 \pm 0.80$ %, respectively, for the SSP5-8.5-SSP2-4.5 difference between 2081 and 2100): the differences in the simulated aerosol spread then change some of the underlying uncertainty, for example in terms of the global mean precipitation response ($-3.79 \pm 0.76$ % for G6sulfur compared

to $-2.07 \pm 0.40$ % for G6solar against SSP2-4.5 between 2081 and 2100). These differences in the aerosols behavior also result in a larger inter-model spread in the regional response in the surface temperatures in the case of the G6sulfur simulations, suggesting the need to devise various, more specific experiments to single out and resolve particular sources of uncertainty. The





spread in the modelled response suggests that a degree of caution is necessary when using these results for assessing specific impacts of geoengineering in various aspects of the Earth system: however, all models agree that, compared to a scenario with

unmitigated warming, stratospheric aerosol geoengineering has the potential to both globally and locally reduce the increase in surface temperatures.

# 1  Introduction

Solar Radiation Modification (SRM) is defined as the proposed artificial altering of the radiative balance of the planet in order to temporarily counteract some of the imbalance produced by the increase in atmospheric greenhouse gases (GHGs). This

might be achieved in multiple ways, but the most studied one, originally proposed by Budyko (1978) and Crutzen (2006) would consist of the injection of $SO_2$ into the stratosphere in order to produce a layer of sulfate aerosols capable of partially reflecting incoming solar radiation: this is usually defined as Stratospheric Aerosol Intervention (SAI), or Sulfate Geoengineering. Simulating such a technique in climate models is the main way of understanding possible impacts to the composition of the atmosphere and to the surface climate, to determine its eventual feasibility, understand its possible impacts on ecosystems and

populations (Zarnetske et al. (2021)) and inform policymakers and stakeholders.

The Geoengineering Model Intercomparison Project (GeoMIP) has been proposed initially in Kravitz et al. (2011) as a way to standardize SRM modeling experiments, allowing for a more robust comparison between model responses and determine sources of uncertainties and areas for improvement. Whereas the term "geoengineering", "climate engineering" or, more re-

cently, "climate intervention" [1] are usually used to consider also methods of Carbon Dioxide Removal (CDR), in the original intention of GeoMIP (and this work) it was only considered as a more colloquial term for SRM.

Two previous experiments in particular have been widely analyzed and discussed: G1, where the solar constant is reduced in order to offset the temperature increase produced by a 4×increase in $CO_2$ compared to pre-industrial concentrations (Kravitz et al. (2013b); Tilmes et al. (2013); Glienke et al. (2015); Russotto and Ackerman (2018b); Kravitz et al. (2020)) and G4, where

a constant amount of $SO_2$ is injected into the equatorial stratosphere, under emissions from the Representative Concentration Pathway 4.5 (RCP4.5) (Pitari et al. (2014); Kashimura et al. (2017); Visioni et al. (2017b); Plazzotta et al. (2019). However, previously performed GeoMIP experiments were not intended to be "realistic" deployments of geoengineering, either because they were performed under idealized conditions (such as 4x$CO_2$ concentrations) or because they considered a fixed, constant amount of injected $SO_2$ with abrupt beginning and ending, and with no baseline simulation to analyse the response against (as

in the case of G4). Two new experiments have been proposed as part of the GeoMIP Phase 6 (Kravitz et al. (2013b)) where geoengineering is aimed at lowering global mean surface temperatures from those in a high-tier emission scenario (Shared Socioeconomic Pathway - SSP5-8.5, Meinshausen et al. (2020)) to those in a medium-tier emission scenario (SSP2-4.5). G6sulfur aims to achieve this temperature goal by increasing the simulated stratospheric aerosol optical depth (AOD): in models with interactive stratospheric aerosol microphysics, this is done by simulating the injection of $SO_2$ between 10°N and 10°S between

---

[1] https://www.silverlining.ngo/us-national-survey-terminology-for-approaches-for-directly-influencing-climate



18 and 20 km, whereas in other models this is done by imposing a sulfate distribution calculated offline. G6solar, on the other hand, decreases total incoming solar irradiance. While the latter does not aim to reproduce the effects of an actual sulfate aerosol intervention, comparisons of its results with simulations of stratospheric aerosols in the same model may help understand the contributions to inter-model differences in the response to aerosols (Niemeier et al. (2013); Visioni et al. (2021)). Both reductions (directly, by turning down the Sun, or indirectly, by having the aerosols reflect the solar radiation) are adjusted

at least every decade to ensure that the target temperature is being met.

    There are multiple uncertainties in the climate models' responses to an artificial, deliberate modification of surface temperatures by means of stratospheric aerosols (Kravitz and MacMartin (2020)) that can be investigated with a multi-model intercomparison. In the stratosphere, these include the conversion of injected $SO_2$ into stratospheric aerosol and the subsequent

large-scale distribution of the aerosols with the stratospheric circulation (not dissimilar to multi-model analyses of simulations of explosive volcanic eruptions, Marshall et al. (2018); Clyne et al. (2020)), the chemical response of key stratospheric components (ozone, methane) to the aerosol layer (Pitari et al. (2014); Visioni et al. (2017b), the magnitude of the produced local heating (Niemeier et al. (2020)), and the dynamical response. At the surface, uncertainties include the magnitude of the resulting global cooling per Tg injected or per unit of optical depth produced, the regional patterns of change in temperature (Kravitz

et al. (2013a)), precipitation (Kravitz et al. (2013b); Tilmes et al. (2013)), and extreme events (Aswathy et al. (2015); Ji et al. (2018)) and other variables that might affect ecosystems and populations (Zarnetske et al. (2021)): for instance, tropospheric ozone (Xia et al. (2017)) or cloud changes (Russotto and Ackerman (2018a)).

    In this work we analyze the response to the two proposed experiments in six global climate models, all part of the Cli-

mate Model Intercomparison Project, Phase 6 (CMIP6). After briefly describing the participating models and the experiment set-ups, in Section 3.1, we first confirm that all models successfully manage to lower globally averaged surface temperatures from those of the underlying high emission scenario to those of the medium one. While in the case of a broad solar reduction there is no constraint on the maximum achievable cooling, previous work has suggested a non-linear behavior between injected $SO_2$ and aerosol burden at high amounts of injections (Pierce et al. (2010); Niemeier and Timmreck (2015)), resulting

in a reduced efficiency. Therefore we also try to evaluate the presence of a similar nonlinearity in the participating models (if it occurs in the range of forcing needed in our experiment). We then analyze in Section 3.2 differences in the latitudinal spread of the stratospheric aerosols cloud despite the consistent injection location. Even when pursuing the same global mean temperature-oriented goal, it has been shown in simulations with CESM1(WACCM) that differences in the latitudinal (Kravitz et al. (2019)) and seasonal (Visioni et al. (2020b)) distribution of the aerosols can result in significant differences in surface

climate. If different models simulate different distribution of the aerosols (as it was for the G4 experiment, Pitari et al. (2014)) due to different (both dynamical and chemical, Niemeier et al. (2020); Franke et al. (2020)) stratospheric processes, the simulated surface climate would also be different. Furthermore, even given similar simulated aerosol distribution, the stratospheric response might differ due to differences in aerosol optics and in the radiative transfer calculation and in the representation of chemical processes in the stratosphere (i.e. if interactive chemistry is considered in the stratosphere, Franke et al. (2020))



resulting in a different dynamical and ultimately surface response (Simpson et al. (2019); Jiang et al. (2019); Banerjee et al. (2020)), which we discuss in Section 3.3 for annual mean temperature and precipitation.

## 2  Description of simulations

We analyze four sets of simulations from 2020 to 2100: two baseline scenarios without geoengineering that follow two Shared Socioeconomic Pathways, SSP2-4.5 and SSP5-8.5 (O'Neill et al. (2016)) and two scenarios with geoengineering, G6solar and
G6sulfur (Kravitz et al. (2015)). Overall, six models participated in all experiments (Table 1).

In the SSP2-4.5 and SSP5-8.5, GHG emissions follow a medium and high trajectory respectively, resulting by the end of the century in a radiative forcing indicated by the last two numbers in the name (i.e., 4.5 and 8.5 W/m$^2$, similar to the Representative Concentration Pathways in CMIP5). The G6 simulations start in 2020 with the same emissions as SSP5-8.5 and, on top of that, have either the solar constant reduced by a certain fraction (in G6solar) or produce a sulfate aerosol optical
depth (in G6sulfur) with the aim of reducing the globally averaged surface temperature down to the SSP2-4.5 level. While the solar reduction is performed in the same way spatially in all G6solar experiments (reducing the solar constant uniformly at all latitudes), not all participating models included stratospheric aerosols by directly injecting SO$_2$. Two models (IPSL-CM6A-LR and UKESM1-0-LL) injected SO$_2$ uniformly between 10°N and 10°S between 18 and 20 km of altitude and across a single longitudinal band (°0). CESM2(WACCM) injected SO$_2$ at the Equator and at 25 km of altitude. The others prescribed an
already-calculated aerosol optical depth distribution: CNRM-ESM2-1 used an input dataset provided by GeoMIP (from the G4SSA experiment Tilmes et al. (2015)), while MPI-ESM prescribed their own aerosol distribution derived from the simulations described in Niemeier and Schmidt (2017); Niemeier et al. (2020). A summary of models participating, ensemble size, and notes related to the implementation of G6sulfur is provided in Table 1.

Two modeling teams, IPSL-CM6A-LR and UKESM1-0-LL, determined every decade how much to reduce the solar constant by or how much more SO$_2$ or prescribed aerosols to have in the stratosphere in order to reduce surface temperatures of the forthcoming decade to SSP2-4.5 levels, whereas four, CESM2(WACCM), MPI-ESM1.2-LR, MPI-ESM1.2-HR and CNRM-ESM2-1, did so every year. For CESM2(WACCM), the determination of injected SO$_2$ or reduction of the solar constant is done by a feedback algorithm described in Kravitz et al. (2017) and also used in Tilmes et al. (2018a, 2020).

## 3  Results

### 3.1  Magnitude of geoengineering required

All models successfully reduce global-mean surface air temperatures to SSP2-4.5 levels to within 0.2°C on average throughout the century with both geoengineering methods (Fig. 1), but the amount of geoengineering required to do so varies across models. There can be a variety of overlapping mechanisms that contribute to these differences. As reported in Table 2, the
models present a large spread in the projected warming produced by the two scenarios. Similar inter-model spreads have been





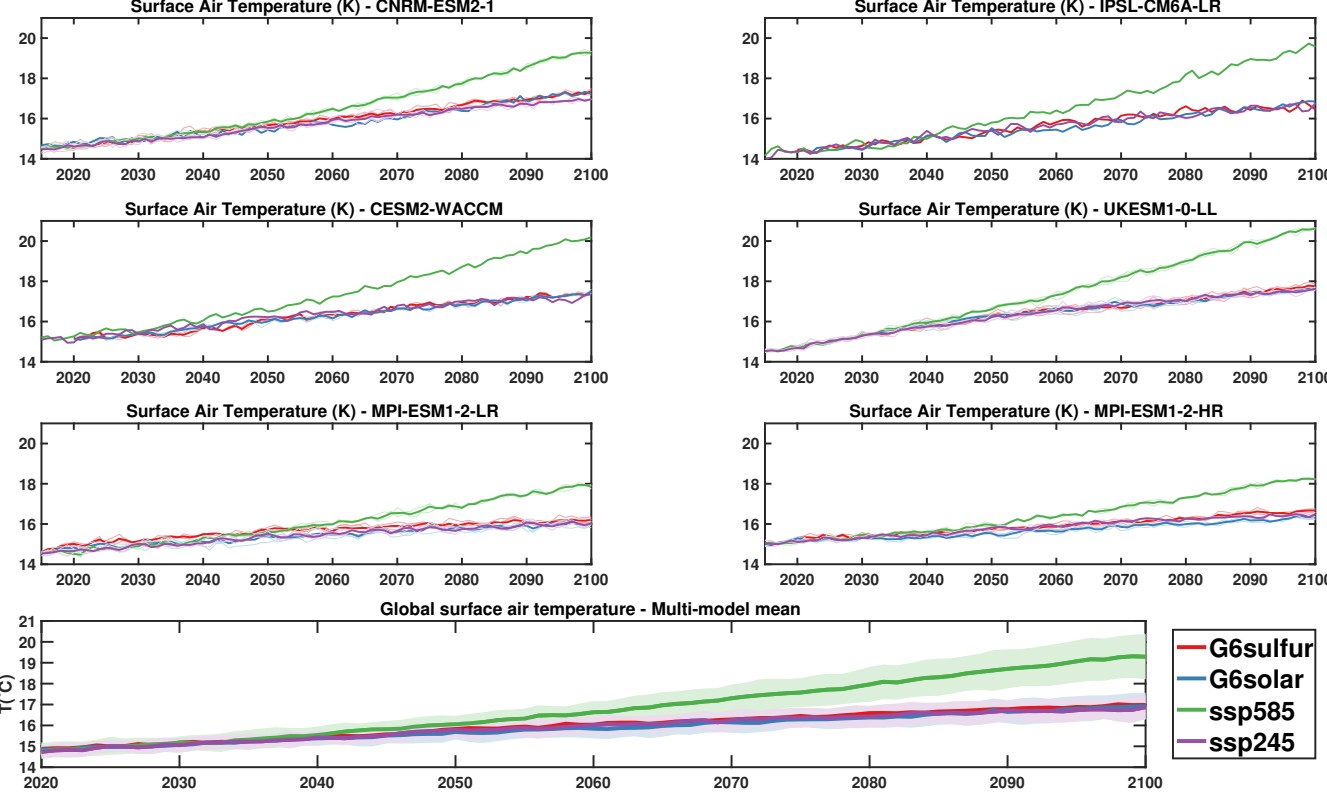

**Figure 1.** Global mean surface temperatures (°) for the four experiments for each participating model. The multi-models mean is shown at the bottom, with the shading representing $1\sigma$ standard deviation of the mean for each experiment.

reported in the recent literature for CMIP6 models for both effective Equilibrium Climate Sensitivity (ECS, the equilibrium warming for a doubling of $CO_2$, see Zelinka et al. (2020)) and Transient Climate Response (TCR, the temperature warming with a doubling of $CO_2$ in a scenario with a 1% per year $CO_2$ increase, see Meehl et al. (2020)), with some models reporting values well above previously established likely ranges for both (Gettelman et al. (2019); Sherwood et al. (2020)). Some of the relationships between the variables reported in Table 2 are explored in Fig. S1: a weak relationship between the different warming in the SSP scenarios and ECS and TCR is to be expected due to differences in both the timescale of the response and the differences in, for instance, other GHGs and tropospheric aerosols (Hansen et al. (2005)) that affect the climate in the short period and that are not factored in the long-term response to $CO_2$ changes. For instance, CNRM-ESM2-1 reported an ECS of 4.79 K (Zelinka et al. (2020)) (the second highest here) but a $\Delta T$ of 1.9 K (the third lowest).

This implies that even if different models agreed on how much either stratospheric AOD or reduction in the solar constant would be needed to cool globally by 1K (the efficacy of the geoengineering method), the overall reported amount of intervention needed would be different due to the different response to the forcing from $CO_2$. To first order, there should be no



expectation that the sensitivity of climate models to a $CO_2$ increase should be related to the reduction in temperature due to

geoengineering (Kravitz et al. (2020)), and we indeed show this in Fig. 2. In Fig. 2e we show that normalizing the required solar dimming or produced AOD to the warming slightly increases the inter-model spread, from 19.9 % to 22.8 % for solar dimming and from 17.2 % to 20.7 % for AOD compared to the mean. In Fig. 2e we also show that the amount of solar reduction and the globally averaged stratospheric AOD seem to be unrelated, suggesting that the mechanisms of cooling by the aerosols and the one related to reduced insolation are different in the analyzed models. For G6sulfur, this might be due not only to the

radiative treatment of the aerosols themselves, but also to different latitudinal distribution in AOD resulting in different forcing, compared to the broad solar reduction that is nearly spatially identical in all models.

The time-dependent amount of geoengineering needed in all models for the two experiments is reported in Fig. 3 (Fig. 3a-b), together with the top-of-atmosphere (TOA) forcing imbalance between SSP5-8.5 and SSP2-4.5, calculated as the outgoing

minus the incoming longwave and shortwave radiation (Fig. 3c), and the underlying difference in $CO_2$ concentration, common to all models, as prescribed for the SSP scenarios in Meinshausen et al. (2020) (Fig. 3d). In terms of TOA forcing, models show a much very consistent forcing that is a result, mostly, of the same $CO_2$ increase, but then they disagree both in the magnitude of the warming produced by this same forcing (as shown in Fig. 1) and in the amount of intervention (optical depth, or solar reduction) needed to overcome that forcing, as shown in panels a and b. The comparison between the two forcing is also

useful to understand the behavior of the geoengineering amount in the models in the first 30 years, where indeed most models indicate little to no geoengineering necessary. CESM2-WACCM is an exception, and indeed shows a slight overcooling in the first decades compared to other models: this is most likely a feature of the current feedback controller, as has been observed in Tilmes et al. (2018a). More in general, the small differences between the two underlying scenarios in terms of global mean temperature in the first decades tend to magnify small differences in the estimated required intervention by the modeling

teams, resulting in larger differences in the first years. Later in the century, when the temperature difference is larger and the intervention scales up, inter-model differences may be explained by the presence of non-linearities or other effects (such as an increase in stratospheric water vapor, Visioni et al. (2017a)). This might explain why all show the same amount of stratospheric AOD in 2050. It is interesting to note that, while a large portion of the models do not vary the amount of geoengineering smoothly, but once a decade, the applied step-function is not evident in the globally averaged surface temperature responses

shown in Fig. 1, where there is no qualitative difference between models in terms of decadal variability: since it is similarly present in the G6solar experiments, the reason for this may be found in the slower oceanic response. Future analyses should investigate whether the step-function introduced by some of the models results in changes in surface climate that, while hidden when considering global or decadal averages, might be present when looking at particular regions or climate features (for instance, the monsoon season) in the years where the step change is present.

## 3.2 Differences in the stratospheric response

For the G6sulfur simulations, the global mean AOD is not, on its own, enough to understand different models behavior. Different spatial distributions of the aerosol layer, while yielding similar global values, might result in different efficiency and would

**Figure 2.** Panels a-e): scatter plot of various relationships between some global quantities in the participating models. ΔT between SSP5-8.5 and SSP2-4.5, global stratospheric aerosol optical depth (SAOD) and solar reduction are defined in the 2081-2100 period. Transient Climate Response (TCR) and effective Equilibrium Climate Sensitivity (ECS) are taken from Zelinka et al. (2020) and Meehl et al. (2020).f) Values in panel c-d) normalized by the ΔT in the same model, to obtain the normalized intervention (green for solar dimming and orange for stratospheric AOD) needed to cool by 1K, with multi-model average on the right and error bars indicating the standard error





**Figure 3.** Time-dependent evolution for all participating models of: a) globally averaged stratospheric AOD increase in the G6sulfur experiment (models with an asterisk in the legend have prescribed AOD); b) solar reduction in the G6solar experiment as a fraction of the overall incoming solar radiation; c) Top-of-atmosphere radiative forcing imbalance (downwelling solar radiation minus upwelling solar+longwave radiation) difference between the two baseline SSP scenarios; d) difference in $CO_2$ concentration between the two emission scenarios from Meinshausen et al. (2020) presented for reference.





produce different responses of the surface climate (MacMartin et al. (2017); Kravitz et al. (2019); Visioni et al. (2020b)). Reasons for a different distribution of the aerosols given similar injection locations of $SO_2$ can be due to different dynamical features of the simulated stratosphere and/or differences in the aerosol microphysics schemes (Pitari et al. (2014); Niemeier et al. (2020); Franke et al. (2020)) resulting in different aerosol growth, transport and sedimentation, as already shown for simulations of explosive volcanic eruptions (Marshall et al. (2018); Clyne et al. (2020)). The response to the presence of the aerosols themselves can in turn produce differences in stratospheric dynamics, for instance interacting with the Quasi-Biennial Oscillation (Aquila et al. (2014); Richter et al. (2017)), strengthening the tropical confinement of the aerosols (Niemeier and Schmidt (2017); Visioni et al. (2018b)). Furthermore, even given similar annually-averaged AOD distributions, differences in the seasonal cycle might lead to different surface climate (Visioni et al. (2020b)). The spatial distributions of AOD for the last decade of the experiment in each model are shown in Fig. 4a. Results vary widely between models: UKESM1-0-LL represents a clear outlier in the tropics, with more than twice the sulfate AOD as other models. At high latitudes, on the other hand, there is a much larger inter-model spread, with values ranging from 0.1 to 0.3 at 90°S and from from 0.2 to 0.45 at 90°N. Strong disagreement between model-simulated AOD in a geoengineering scenario was already reported in Pitari et al. (2014) and Plazzotta et al. (2018) for the G4 experiments, where a 5 Tg-$SO_2$/yr injection in the equatorial stratosphere was prescribed in the simulation protocols. No models used in that experiment have been used in the G6 scenarios, so a direct comparison can't be done with different versions of the same models. In this case, however, we can note that all models at least agree on the presence of a confinement of a portion of the aerosols in the tropical pipe, whereas in G4 half of the models reported much less AOD in the tropics and more at very high latitudes (Pitari et al. (2014)), which is physically very unlikely given observations from the Pinatubo eruption in 1991 (Robock (2000); Pitari et al. (2016)).

Model spread, of course, is not the same as uncertainty and might either be smaller (models agree despite lack of observational support) or larger (if some model results are simply inconsistent with available observations). Here we try to better constrain the distribution of AOD in the various models in G6sulfur using the up-to-date CMIP6 dataset for volcanic forcing, that combines measurements from various sources (Dhomse et al. (2020), retrieved from ftp://iacftp.ethz.ch/pub_read/luo/CMIP6/, last access: October 29, 2020). In particular, using the 550 nm extinction data, we derive the stratosphere-only latitudinal distribution of the optical depth following the Pinatubo 1991 eruption, averaged from 1 month after the eruption (July 1991) to 1 year after, in order to also consider the poleward transport of the aerosols. It needs to be highlighted that the comparison between an impulsive injection (as Pinatubo) versus a sustained injection (as in the geoengineering experiment) is an imperfect one, both in terms of the aerosol distribution and in terms of the effects on surface climate (Duan et al. (2019)), but it is possibly the only "real" point of comparison between model behavior and the actual atmospheric behavior. In Fig. 4c we report the AOD from Pinatubo thus derived and compare the results with those from the various G6sulfur models considering the year in which each model reaches the same global value of AOD. This comparison highlights various elements that would be lost considering the results towards the end of the century as in Fig. 4a: models show a higher agreement considering a moderate level of global AOD reached, and compared with the results from Pinatubo (considering the differences in meteorology and injection location) they look reasonable. In particular, UKESM1-0-LL and CESM2-WACCM show a better agreement in their tropical AOD, as





opposed to what was shown in Fig.4a, indicating the presence of non-linearities at high injection rates, that might be induced
in UKESM1-0-LL by a too strong confinement of the aerosols in the tropical pipe as a consequence of the dynamic response
to heating (Aquila et al. (2014); Niemeier and Schmidt (2017); Visioni et al. (2018b). In Fig. 4c models show a much better
agreement also at high latitudes (at least in the northern hemisphere) compared to Fig. 4a, with the exception of the prescribed
AOD in CNRM-ESM2-1, again pointing, when comparing with Fig.4a, to the stronger interaction of dynamical changes with
the simulated AOD also at high latitudes when considering higher injection loads (Visioni et al. (2020a)).

The amount of SO$_2$ needed to reach a certain stratospheric AOD varies considerably between climate models with interactive
stratospheric aerosols even for simulations of Pinatubo, ranging in current estimates between 10 and 20 Tg-SO$_2$ with a central
value of 14 (Timmreck et al. (2018)). In the G6sulfur experiments, the models show discrepancies in the estimate of the amount
needed to achieve a similar global AOD as in Pinatubo (with a multi-model average of $9.3 \pm 2.3$ Tg-SO$_2$, see table in Fig. 4) ,
closer to the lower limit from Timmreck et al. (2018) (10 Tg-SO$_2$) for UKESM1-0-LL and IPSL-CM6A-LR and 60% lower for
CESM2-WACCM. For CESM2-WACCM, the difference could be partially explained by the difference in altitude for the SO$_2$
injections. In Fig.4c we also report the cooling produced by the G6sulfur aerosols, compared to SSP5-8.5 in the considered year
(we used a 5-years average around that year to reduce the contribution of natural variablity). For Pinatubo, there is uncertainty
in the cooling produced by the volcanic aerosols due to the precise meteorology of that year (for instance, the influence of an
El-Niño event or other climatic oscillations compared to the years immediately before/after): Parker et al. (1996) estimate a
global cooling of around 0.4 K, and similarly Soden et al. (2002) estimated a range between 0.3 and 0.5 K. The multi-model
average for the G6sulfur simulation is very similar, at $0.46K \pm 0.09$, but there is a large range in the single values from 0.24
(in MPI-ESM1-2-LR) to 0.74 (for CESM2-WACCM). Overall, the comparisons shown in Fig. 4 raise an important point that
should be taken into account when analysing G6 simulations in future works: while limiting the analyses towards the end of
the century might yield a higher signal-to-noise ratio, it also risks magnifying uncertainties related to non-linear processes in
the stratosphere. In Fig. S1, we also report the yearly evolution of the latitudinal distribution of AOD for models that inject
SO$_2$, normalized by the amount of SO$_2$ injected in that year, which clearly shows the decrease in efficiency at higher injection
loads.

As mentioned before, the presence of the aerosols in the stratosphere also produces a perturbation of stratospheric dynamics
(Richter et al. (2017); Visioni et al. (2020a)) that, in turn, might affect precipitation (Simpson et al. (2019)) and temperature
(Jiang et al. (2019)) at the surface. The response is driven by the absorption of infrared radiation by the aerosols resulting
in the heating of the stratospheric air, and is thus dependent on the overall burden and the size of the particles (Pitari et al.
(2016)), but also on interactions with the chemical cycles in the stratosphere (Visioni et al. (2017b); Richter et al. (2017)) and
the incursion of water vapor from the troposphere due to the warming of the tropopause layer (Visioni et al. (2017b); Tilmes
et al. (2018b); Boucher et al. (2017)). In Fig. 5 we show the stratospheric temperatures in the last decade of the G6sulfur
experiment for all models. Interestingly, the model with the highest AOD in the tropics, UKESM1-0-LL, is also the model
showing the least amount of stratospheric heating, whereas IPSL-CM6A-LR, with an average tropical AOD (but much larger





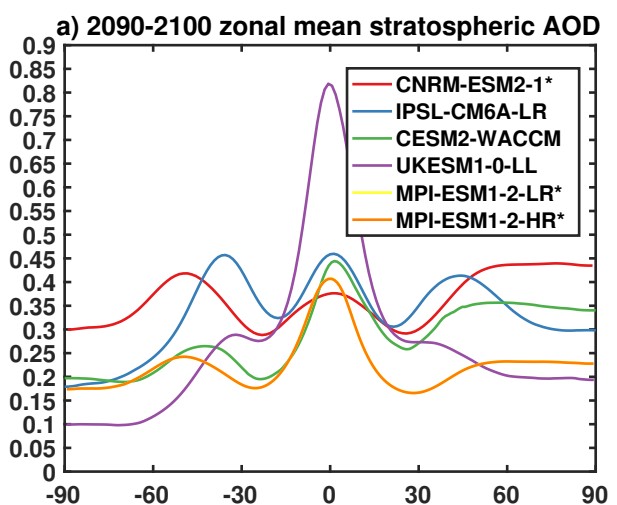

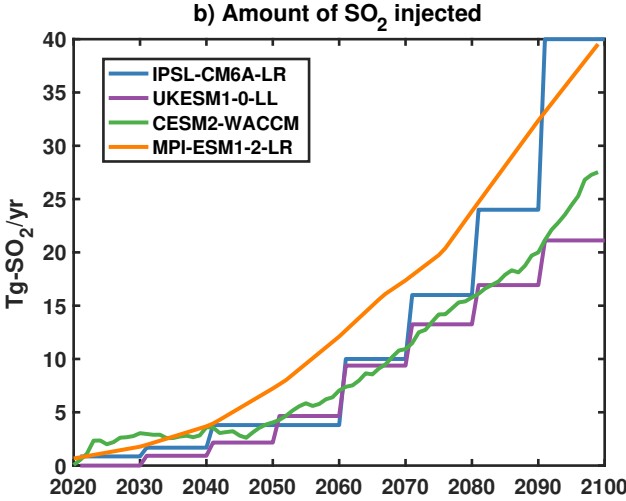

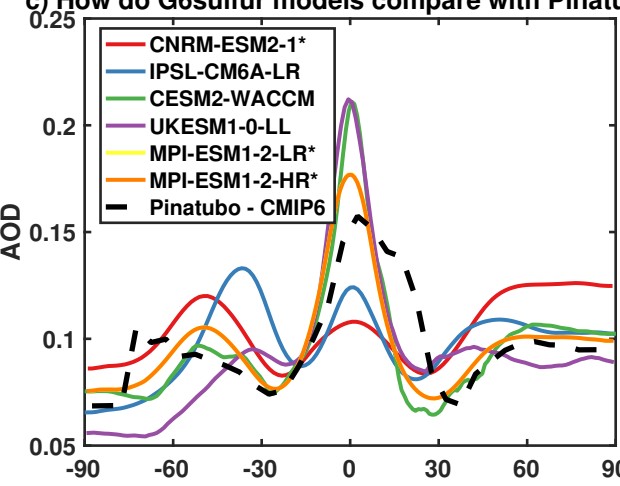

**Figure 4.** a) Stratospheric AOD in the last decade of the experiment for all participating models. The asterisk in the legend indicates models with prescribed optical depth. b) Injected $SO_2$ for available models, in Tg-$SO_2$/yr. c) AOD distribution for each model in the year with a global AOD closer to that from Pinatubo (0.102, averaged from July 1991 to June 1992), and comparison with the latitudinal distribution for the volcanic eruption following the new CMIP6 composed dataset (Dhomse et al. (2020)). In the box to the right of panel c), the year where the global value of AOD reaches 0.102 in the model is indicated, together with the amount of $SO_2$ needed to achieve that value and the cooling produced in G6sulfur compared to SSP5-8.5 in that year. Models marked with an asterisk in the legend used prescribed aerosol distributions for G6sulfur.





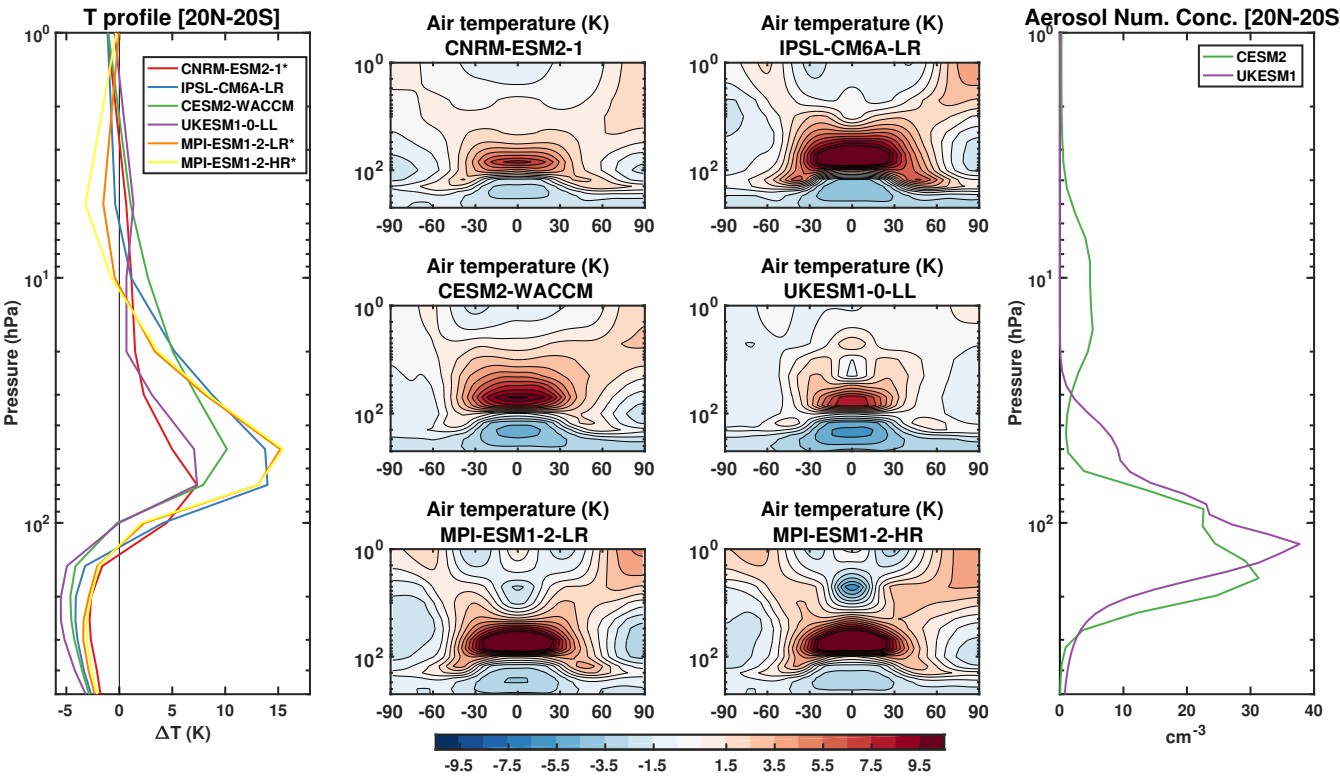

**Figure 5.** Profile of stratospheric temperatures changes (G6sulfur-SSP2-4.5) between 20°N and 20°S are shown in the left panel. In the central panels, the changes are shown for each participating model. Profile of aerosol number concentration are shown in the right panel for a select number of models where output was available. All changes are for the years 2091-2100, and evaluated against the same period for the underlying emission scenario SSP5-8.5)

SO$_2$ injection needed to achieve it) shows a temperature change that is much larger than the other models. The reasons for this may depend on multiple aspects that would need to investigated separately: for instance these might be a different size distribution of the stratospheric aerosols or a different concentration of particles (shown in Fig. 5) differences in ozone changes resulting in different heating rates (Richter et al. (2017); Niemeier et al. (2020), heating from stratospheric water vapor (Pitari et al. (2014); Simpson et al. (2019) or differences in the radiative schemes between models.

### 3.3 Surface climate response

When geoengineering the climate, reducing incoming solar radiation (either simulating stratospheric aerosols, or by reducing the solar constant in models) to obtain the same global surface temperature as a scenario with lower GHGs does not assure that regional temperatures follow the same pattern. This has been reported in climate model simulations of various complexity, from 1-D models (Henry and Merlis (2020)) to Earth System Model simulations (i.e., Ban-Weiss and Caldeira (2010); Niemeier et al. (2013); Jones et al. (2018); Visioni et al. (2021)). These differences may be reduced if, together with reducing global





temperatures, the geoengineering strategy aims to also reduce differences in higher-order temperature gradients (Kravitz et al.
(2016); Tilmes et al. (2018a), but they cannot be completely cancelled due to various factors. First and foremost, a fundamental
differences in the radiative fluxes from $CO_2$ (that warm throughout the atmospheric column) and from the reduction in solar
constant (that cool from the bottom-up) (Ban-Weiss and Caldeira (2010); Henry and Merlis (2020)) and from their seasonal
and latitudinal differences (Govindasamy et al. (2003); Ban-Weiss and Caldeira (2010); Visioni et al. (2020b)) and surface
climate effects (such as precipitation changes) of the stratospheric heating produced by the aerosols (Simpson et al. (2019);
Visioni et al. (2021); Jones et al. (2021)). Other factors may also be surface effects of the stratospheric heating produced by
the aerosols (Simpson et al. (2019); Banerjee et al. (2020) and an inability to restore the same state for the ocean circulation:
this latter point has been observed for instance in CESM1(WACCM) in Fasullo et al. (2018), and in one of the models that
performed G6 simulations, CESM2(WACCM) in Tilmes et al. (2020).

All of these differences are compounded with those already present in climate models for regional temperature projections
for $CO_2$ increases: on this point, however, MacMartin et al. (2015) argued that reducing surface temperatures through geo-
engineering has the potential to actually reduce model spread in regional projections. That work however considered the G1
experiment, that entails a uniform solar reduction to reduce temperatures under a $4\times CO_2$ increase. Clearly then, most of the
differences listed above are not included in such an idealized experiment. This is clear when looking at the multi-model aver-
ages of surface temperature differences shown in Fig.6: not only are the simulated differences with SSP2-4.5 much larger in
G6sulfur compared to G6solar, but the inter-model spread is much smaller in G6solar, showing better agreement between mod-
els when the uncertainties related to the stratospheric sulfate are removed. For G6sulfur models, there is a general agreement in
the inability of sulfate geoengineering to cool down the northern high-latitudes, partly due to the focus of the geoengineering
strategy on reducing global mean temperatures (Kravitz et al. (2019)), but also due to the presence of stratospheric heating
(Jiang et al. (2019)), as evident by the absence of a surface warming of the same magnitude in the G6solar simulations. The
residual warming present also in the G6solar simulations can be explained by the differences in the radiative forcing from
the $CO_2$ and the solar reduction (Ban-Weiss and Caldeira (2010); Henry and Merlis (2020); Visioni et al. (2021)). Differ-
ences in the surface response between models would thus depend on how different models physically reproduce some of the
processes mentioned, but also on the differences in the stratospheric response reported in the previous section: different lat-
itudinal and seasonal distributions of the aerosols produce different climate states even just in the same model (as shown in
CESM1(WACCM) in Kravitz et al. (2019); Visioni et al. (2020b)), and the stratospheric heating is also reportedly different as
shown in Fig. 5. Nonetheless, the essential finding from MacMartin et al. (2015) still holds when comparing the multi-model
standard error for the geoengineering projections against those for the SSP5-8.5 changes, that especially over land and at high
latitudes are always higher than both G6 cases.

We report the surface temperature maps for the last two decades of the experiment for each model in Fig. 7: from them, some
observations can be made that would not be immediately evident from the multi-model average. For G6sulfur, there is a good
agreement regarding the residual warming over Northern Eurasia across models, with the exception of CESM2 (that is bal-
anced in the multi-model average by a stronger warming modeled by MPI-ESM). There is less agreement over North America,

**Figure 6.** Left column: multi-model averages for surface temperatures changes averaged over 2081-2100 in different cases (a) SSP5-8.5; c) G6sulfur; e) G6solar) minus the same period for SSP2-4.5. Etched areas (in grey) indicate where less than 66% of models (here, 4 out of 6) agree on the sign of the difference in that grid-point. Right column: Standard error in the multi-model mean for the same reference case on the left. All models results have been re-gridded using a common grid equivalent to that from the model with the lowest horizontal resolution.





where some models simulate a cooling in G6sulfur compared to SSP2-4.5 while some simulate a warming. This might be due

to differences in the response of the North Atlantic circulation both to increasing GHGs and to geoengineering (Tilmes et al. (2018a, 2020)). Comparing this result to that from G6solar, where there is a concurrence of all models in simulating a small warming over the same region, might indicate that the much different response in G6sulfur might on the other hand be due to differences in the distribution of the stratospheric aerosols: UKESM1-0-LL, for instance, where more residual warming is present, shows the lowest AOD over high latitudes (Fig. 4). In the tropics, in the Amazon region models seem to differ more in

the G6sulfur case and less in the G6solar case: possible causes might be an influence from the different magnitude of AOD in that region, different responses of the vegetation to increasing $CO_2$ concentrations and reduced solar radiation (Simpson et al. (2019)) or local changes in atmospheric circulation (Jones et al. (2018)).

Overall, the inter-model differences indicate the need for some care when trying to understand the possible surface impacts

of sulfate geoengineering by using multi-model ensembles. It might be difficult to correctly separate the differences in surface impacts due to differences in the stratospheric AOD (shown in Fig.4) given a similar injection, and those produced by different response of the surface climate. While comparing results with those from a similar, more uniform experimental design such as G6solar might help, the lack of the potential response produced by the aerosols (Banerjee et al. (2020); Visioni et al. (2021)) may suggest the use of a prescribed aerosol distribution for various models (Tilmes et al. (2015)) as an intermediate approach.

This can also be seen in the comparison between the two version of MPI (that differ only in their horizontal resolution, which is twice as high in the HR version): they both use the same AOD distribution, and have the same magnitude of stratospheric AOD in the whole period. Yet, they show some considerable differences in the surface temperature response to the same aerosol (or even solar) forcing. In particular, the warming observed over North America in the LR version is not present in the HR version, whereas the warming present in West Antarctica in the HR version is not present in the LR version. This might indicate that

regionally the temperature response may be due to different response of the deep ocean circulation (in the West Antarctica case) as also shown in McCusker et al. (2015), and that this might be model dependant (other than depending on the particular injection strategy); or to a different response of the atmospheric circulation (Jones et al. (2018)). On the other hand parts of the response, such as the patches of warming present in the Amazon and in Central Africa, possibly due to a different land response, are shared between the two versions, and similarly a large part of the warming over Eurasia. While observing the response of

different versions of the same model to the same forcing might point out to some of the causes, comparing that to the response of a different model to the same forcing may also highlight which parts of the overall response is model-dependant, and which is robust across models.

Surface temperatures are not the only measure of the possible impacts of either climate change or geoengineering: amongst the many others, hydrological cycle changes are also central to any assessment. Under climate change, due to the surface

and tropospheric warming allowing for more moisture to be retained by the air, global precipitation has been consistently projected to increase (Pendergrass and Hartmann (2014)), and a similar behavior is displayed by the models participating in the G6 experiments (8). Similarly, it has been widely assessed that trying to restore surface temperature to a previous state by means of modifying the top of the atmosphere radiative balance tends to overcompensate the changes in precipitation, therefore



**Figure 7.** Surface temperatures changes in the period 2081-2100 in G6sulfur compared to the same period for SSP2-4.5 in G6sulfur simulations (left panels) and G6solar simulations (right panels) for all participating models. Shaded areas indicate where the difference is not statistically significant, evaluated using a double-sided t-test with p<0.05 on the ensemble averages for each model.





reducing global mean precipitation. Globally, the changes are driven by the perturbation of the surface heat fluxes (Tilmes et al.
(2013); Kravitz et al. (2013b); Niemeier et al. (2013)) and changes in sea-land temperature contrast: regionally however, the modification of the baseline distribution of precipitation can be due to changes in the Inter-tropical Convergence Zone (ITCZ, Russotto and Ackerman (2018b); Cheng et al. (2019)) produced by changes in the inter-hemispheric temperature gradient, general circulation changes produced by stratospheric heating (Simpson et al. (2019)) and regional and seasonal changes in heat fluxes and temperature gradients (Jones et al. (2018); Visioni et al. (2020b)). In the case of sulfate injections, these changes
can be strongly dependent on latitudinal and temporal distribution of the aerosol cloud as well (Kravitz et al. (2019); Visioni et al. (2020b)).

The response of the various models for the G6 experiments in Fig. 8 is in agreement that the global-mean precipitation would be overcompensated (Niemeier et al. (2013)). However, models disagree on the magnitude of this overcompensation, and in the difference between G6solar and G6sulfur. The fact that under the SSP2-4.5 scenario some warming continues during the
$21^{st}$ century, combined with the precipitation overcompensation by geoengineering, results in some models in no changes in global precipitation compared to the beginning of the century (as already noted in Irvine and Keith (2020)): only G6sulfur in IPSL-CM6A-LR shows a decrease compared to that period by the end of the century. For the purpose of future analyses, the anomalous global precipitation response in the MPI models for G6sulfur has to be noted: it is very likely that the slightly larger response in global mean precipitation at the beginning of the century is due to differences in the initialization process for those
simulations, rather than in a change produced by the sulfate (which is very close to zero, in 2020), and results before 2050 (for the LR version) or 2040 (for the HR version) should not be considered as representative.

From the prospect of assessing ecosystem impacts, this decoupling of precipitation, temperatures and $CO_2$ should be investigated in depth to understand if and where it would be beneficial or not, and it further stresses the notion that reducing
precipitation is not an automatic result of geoengineering, but that the outcome is related to which specific cooling targets geoengineering is deployed to achieve (Tilmes et al. (2013); Irvine et al. (2019); Lee et al. (2020)). All models agree that global precipitation changes under G6sulfur are larger than the same changes under G6solar: there might be various reasons for this, such as differences in latent heat due to different ratios of diffuse solar radiation (that increases in the case of the sulfate aerosols, Visioni et al. (2021)) resulting in more atmospheric absorption, changes in cloud formation produced by the
different vertical atmospheric temperature gradient. Niemeier et al. (2013) suggested that the reason for this might be found in the stratospheric heating produced by the aerosols resulting in more water vapor entering the stratosphere from the warming of the tropopause layer (Tilmes et al. (2018a); Simpson et al. (2019)) producing a small positive radiative forcing whose warming effect (Hansen et al. (2005); Visioni et al. (2017a)) needs to be counterbalanced by injecting slightly more aerosols.

Lastly, model agreement regarding regional changes tend to be lower in G6sulfur than in G6solar (Fig. 9), but all models project most of the significant changes observed over the tropics (where also most of the baseline precipitation is located), but with some significant local differences between models (Fig. 10): for instance, while CESM2-WACCM shows less precipitation in the tropical northern hemisphere and more precipitation in the tropical southern hemisphere, UKESM1-0-LL presents a





drying in both hemispheres, especially over continents. In some cases, such as at high northern latitudes, all models show a
positive change in G6sulfur, and a negative change in G6solar. It is again interesting to note the differences in the projected
precipitation changes in the two version of MPI: the HR version shows a further decrease in precipitation in the tropics
compared to the LR version, and at high latitudes LR shows much higher changes compared to HR. This shows that even
given the same AOD distribution, and similar models, some of the observed changes in the case of SAI may differ depending
on the simulated response of the circulation to the same forcing. In this work we have only analyzed the annual response to
precipitation, but there are many regions where changes to the seasonal cycle of precipitation may be even more crucial, such
as those that experience a monsoon climate, and whose cycle might be affected by SAI (see for instance Simpson et al. (2019);
Visioni et al. (2020b) for the Indian subcontinent, and Da-Allada et al. (2020) for Western Africa): an in depth analyses of
these impacts would also be necessary. Interestingly, unlike for the multi-model mean for surface temperatures (Fig. 6), the
multi-model standard error for precipitation is very similar and in some cases higher in G6sulfur than in SSP5-8.5, indicating
that, while true that reducing surface temperatures would indeed reduce disagreement in future projections between models,
that might not hold true for other impacts (of which precipitation might only be an example), where due to the influence of
changes in surface temperatures, effects driven by $CO_2$ and possible changes in dynamical changes driven by the aerosols,
modeling uncertainties might remain higher either with high $CO_2$ or with geoengineering.





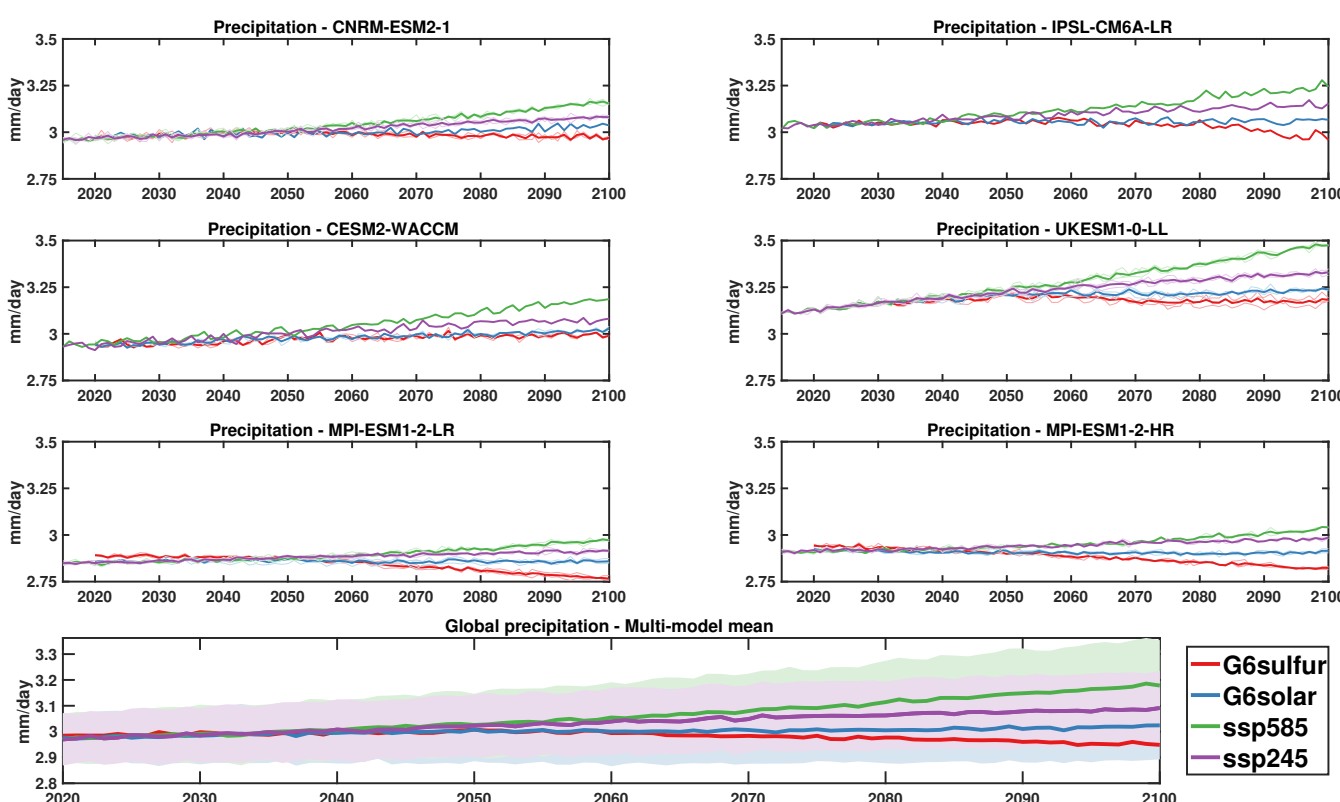

**Figure 8.** Global mean precipitation (mm/day) for the four experiments for each participating model. The multi-models mean is shown at the bottom, with the shading representing $1\sigma$ standard deviation of the mean for each experiment.





**Table 1.** Summary of model simulations used in this work. The first column has the name of the model used, the doi for the relative CMIP6 dataset as recommended by CMIP6 (see Stockhause and Lautenschlager (2021)) and the horizontal and vertical resolution, whereas column 2 indicates the main scientific reference where the model version is described. Columns 3 to 6 show the size of the ensemble analyzed in this work: for some models, more ensemble members are available for the SSP experiments, but only those with the same variant as the G6 experiments are used in this work. Finally, the last two columns indicate the source of stratospheric aerosols for G6 and the presence of interactive stratospheric ozone.

| Model name and CMIP6 doi (resolution†) | Main scientific reference(s) | SSP2-4.5 | SSP5-8.5 | G6solar | G6sulfur | Stratospheric aerosols in G6sulfur | Interactive stratospheric ozone |
|---|---|---|---|---|---|---|---|
| | | (number of simulations and variant names) | | | | | |
| CESM2(WACCM) Danabasoglu (2019) h:288×192,v:49 | Danabasoglu et al. (2020) | 2 r1,r2 | 2 r1,r2 | 2 r1,r2 | 2 r1,r2 | From SO$_2$ injection‡ | Yes |
| CNRM-ESM2-1 Seferian (2018) h:256×128,v:40 | Séférian et al. (2019) | 3 r1,r2,r3 | 3 r1 | 1 r1 | 3 r1,r2,r3 | AOD scaled from Tilmes et al. (2015) | Yes Michou et al. (2020) |
| IPSL-CM6A-LR Boucher et al. (2018) h:144×143,v:79 | Boucher et al. (2020) Lurton et al. (2020) | 1 r1 | 1 r1 | 1 r1 | 1 r1 | From SO$_2$ injection | No |
| MPI-ESM1.2-LR Wieners et al. (2019) h:192×96,v:47 | Muller et al. (2018) | 3 r1,r2,r3 | 3 r1,r2,r | 3 r1,r2,r | 3 r1,r2,r | AOD scaled from Niemeier and Schmidt (2017) | No |
| MPI-ESM1.2-HR Jungclaus et al. (2019) h:384×192,v:95 | Muller et al. (2018) | 3 r1,r2,r3 | 3 r1,r2,r | 3 r1,r2,r | 3 r1,r2,r | AOD scaled from Niemeier and Schmidt (2017) | No |
| UKESM1-0-LL Tang et al. (2019) h:192×144,v:85 | Sellar et al. (2019) | 3 r1,r4,r8 | 3 r1,r4,r8 | 3 r1,r4,r8 | 3 r1,r4,r8 | From SO$_2$ injection | Yes |

† resolution is described as horizontal (h), lat × lon, and vertical (v). ‡Injected at the Equator at 25 km in deviation from the protocol described by Kravitz et al. (2015)



**Table 2.** Summary of results for the simulations in this work for the last decade of the experiment (2081-2100). When applicable, values are considered as global mean, ensemble mean averages. In the last three column, the solar reduction needed in the G1 experiment (Kravitz et al. (2020)) to offset the forcing of a $4\times CO_2$ increase, the effective Equilibrium Climate Sensitivity (ECS) from Zelinka et al. (2020) and the Transient Climate Response (TCR) from Meehl et al. (2020) are included for comparison.

| Model | $\Delta T$ (K) (SSP58.5-SSP24.5) | AOD | Injected $SO_2$ (Tg-$SO_2$/yr) | Solar Reduction (G6) (%) | Solar Reduction (G1) (%, Kravitz et al. (2020)) | ECS (K) Zelinka et al. (2020) | TCR (K) |
|---|---|---|---|---|---|---|---|
| CESM2(WACCM) | 2.42 | 0.296 | 21[†] | 2.33 | 4.91 | 4.68 | 2.0 |
| CNRM-ESM2-1 | 1.90 | 0.327 | N.A.[‡] | 1.36 | 3.72 | 4.79 | 1.9 |
| IPSL-CM6A-LR | 2.40 | 0.363 | 40[†] | 1.78 | 4.10 | 4.56 | 2.3 |
| MPI-ESM1.2-LR | 1.58 | 0.235 | 36[‡] | 2.15 | 4.57 | 2.98 | 1.8 |
| MPI-ESM1.2-HR | 1.50 | 0.235 | 36[‡] | 2.15 | N.A. | 2.98 | 1.7 |
| UKESM1-0-LL | 2.54 | 0.357 | 21[†] | 2.18 | 3.80 | 5.36 | 2.8 |
| Model average | $2.05 \pm 0.42$ K | $0.307 \pm 0.061$ | $29 \pm 9$ | $1.99 \pm 0.36$ | $4.22 \pm 1.00$ | $2.1 \pm 0.4$ | |

[†] Models using emissions of $SO_2$. [‡] Models using prescribed AOD.

**Figure 9.** Left column: multi-model averages for precipitation changes averaged over 2081-2100 in different cases (a) SSP5-8.5; c) G6sulfur; e) G6solar) minus the same period for SSP2-4.5. Etched areas (in grey) indicate where less than 66% of models (here, 4 out of 6) agree on the sign of the difference in that grid-point. Right column: Standard error in the multi-model mean for the same reference case on the left. All models results have been re-gridded using a common grid equivalent to that from the model with the lowest horizontal resolution.





**Figure 10.** Precipitation changes (mm/day) in the period 2081-2100 in G6sulfur compared to the same period for SSP2-4.5 in G6sulfur simulations and G6solar simulations(left panels) for all participating models. Shaded areas indicate where the difference is not statistically significant, evaluated using a double-sided t-test with p<0.05. On the left we show the zonal mean values for SSP2-4.5 (straight lines) and SSP5-8.5 (dotted lines) while on the right we show the % changes in the two geoengineering cases.





# 4 Conclusions

We have shown in this work some preliminary results from the G6sulfur and G6solar modeling experiments, proposed in Kravitz et al. (2015) for the Geoengineering Model Intercomparison Project, part of the Climate Model Intercomparison Project Phase 6. These two new experiments aim to reduce global temperatures in the $21^{st}$ century from those simulated under a high-tier emissions scenario (SSP5-8.5) to those simulated under a medium-tier emissions scenario (SSP2-4.5), either by simulating the artificial injection of stratospheric aerosol precursors in the stratosphere, or by reducing the solar constant in the models. In terms of surface climate response, some broad features are shared by all models, such as a reduction in global mean precipitation and a residual warming in the northern high latitudes (Henry and Merlis (2020)), particularly present in G6sulfur (Simpson et al. (2019); Banerjee et al. (2020)). Other locations show more disagreements between models in terms of the surface temperature response: the larger uniformity in the response between G6solar simulations, where the solar dimming is applied in the same latitudinally-uniform way in all models, suggests that part of the surface response uncertainty in G6sulfur is driven by differences in the latitudinal distribution of the aerosols and not to a different response of the surface climate to the same radiative forcing.

The comparison of the two experiments may help in various ways: when comparing the single-model response to the two different forcings, it helps highlight some of the physical differences between the two interventions (as in Visioni et al. (2021)), produced by the the stratospheric aerosols physical and chemical effects. Analysing the inter-model spread also highlights the degree to which uncertainty in surface climate response to stratospheric aerosols is driven by uncertainties in the stratospheric processes, versus uncertainties in how the climate response to a specified forcing such as reduced insolation, and may point to a path to successfully identify and, eventually, reduce some of them. We have shown that large inter-model variability remains in the distribution of the aerosol after injections of $SO_2$ in the tropical stratosphere, as well as in the temperature response of the stratospheric air. As we discussed in Section 3.2, the resulting latitudinal distribution of the aerosols given similar injection locations can be due to multiple factors; in particular stratospheric dynamics differences regulating the large-scale transport of the aerosols and the microphysical differences regulating the oxidation of $SO_2$ and the subsequent growth of the aerosols. The interaction between the stratospheric aerosols and the rest of the system further complicates the identification of a single mechanism by which to aerosol distributions might differ: there may be uncertainties related to the simulated radiative inter- action (for instance, the rate of absorption of IR radiation by the aerosols) and stratospheric chemistry (i.e., changes in ozone chemistry, which in turns affect local radiative transfer) that may produce different localized heating of air and thus affect dif- ferently both the surface climate and stratospheric dynamics (which, in turn, may affect the aerosol distribution, Niemeier and Schmidt (2017); Kleinschmitt et al. (2018)). All these uncertainties in stratospheric dynamics (summarized in Fig.11) can thus indirectly affect surface climate in simulations of geoengineering with stratospheric aerosols, by means of a different reflection of sunlight depending on the resulting distribution of the aerosols. This type of uncertainty is thus separated from those directly connected to a stratospheric influence on various aspects of the surface climate: local surface temperatures (Jiang et al. (2019)),



precipitation (Simpson et al. (2019)) or cloud cover changes (Visioni et al. (2018a)).

Simulations such as those we analyzed here can give useful information on the current range of uncertainty over many
projected impacts of geoengineering. In particular, the successful coupling of the new Earth System Models used in CMIP6
with land, ocean and cryosphere components can help with the exploration of various impacts, for instance on ecosystems
(Zarnetske et al., 2021) or ice sheets melting (Fettweis et al. (2020)), which are crucial to properly inform policymakers and
interested parties, and the inter-model spread can help in communicating the uncertainties tied to those projections. As we
outlined above, however, these simulations may not be as useful in helping reduce most of these uncertainties: it is therefore
important not to rely only on these simulations going forward, but to devise new experiments that might improve the accuracy
with which we model the relevant interactions in the atmosphere. To do so, there may be multiple venues: one way could
be using different physical-based approaches to modeling that don't involve 3D climate modeling and that might shed light
on the single processes (i.e. for instance Dai et al. (2018); Lutsko et al. (2020); Seeley et al. (2021), or plume modeling), lab
experiments trying to replicate the conditions of the stratosphere (Dai et al. (2020)). Another way could be using global climate
models but trying to constrain some of the various processes in order to reduce uncertainty: this could be done, for instance, by
prescribing the same stratospheric aerosol distribution in different models (as suggested in Tilmes et al. (2015)) and as some
models do in this work, or modifying some parameters in the model simulation while keeping everything else fixed to constrain
a source of uncertainty (as proposed for volcanic eruption by Timmreck et al. (2018) in the Pinatubo Emulation in Multiple
models (PoEMs) experiment), or by continuing to simulate a constant solar dimming in place of the more complex aerosols
(see for instance Irvine et al. (2019)) to understand portions of the global surface response. All of these (and more) methods
combined may be able to increase our confidence when projecting the impacts of sulfate geoengineering as a short-term addi-
tion to mitigation (but not as its replacement, MacMartin et al. (2018); de Coninck et al. (2018)) in order to limit the harmful
impacts of climate change.

When considering the possible impacts of SAI using GeoMIP simulations, it should also be considered that the injection
strategy simulated in the G6 experiments is only one of the possible ways in which SAI could be deployed, and for various
reasons, it is not even the most ideal one. Kravitz et al. (2019) showed that a strategy that makes use of different locations of
injection outside the equator (MacMartin et al. (2017)) in order to manage not just global mean temperatures, but also inter-
hemispheric and equator-to-pole temperature gradients, would further reduce harmful impacts by better restoring sea-ice and
restore the ITCZ. Further, injecting all days of the year might also not be the most ideal choice (Visioni et al. (2019)), and
some of the resulting climatic effects might depend on the seasonal distribution of the aerosol cloud (Visioni et al. (2020b)).
So, while the coordinated experiment described in this work might be good as a starting point, it should not be considered
as the only way in which SAI might be deployed. This is also valid in terms of the underlying emission scenario used, as
a future where emissions continue unabated is not the ideal one in which an eventual SAI deployment should be imagined,
even if it might mitigate the short-term effects of the GHG-induced warming. A scenario where emissions are cut, but not fast
enough, and global temperature thresholds set by international agreements may be temporarily exceeded could be one where





## Main physical sources of uncertainties for Stratospheric Aerosol Intervention

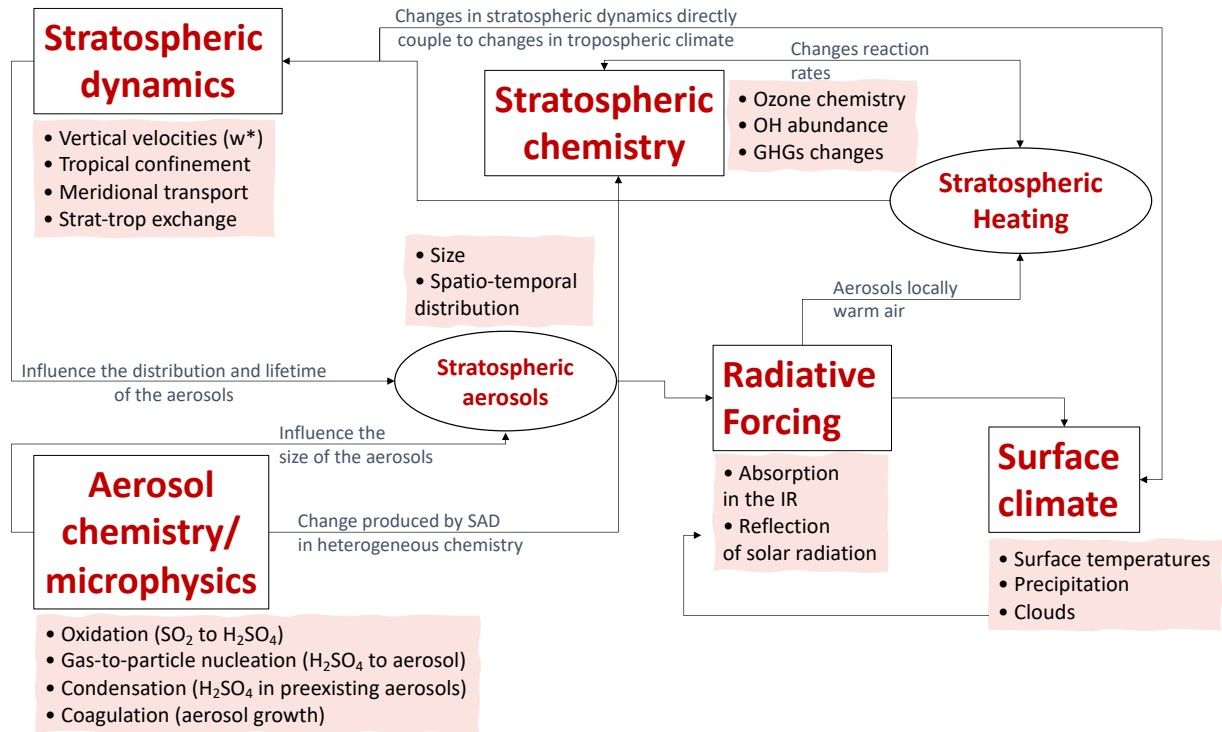

**Figure 11.** Scheme exemplifying the sources of uncertainties in modeling stratospheric aerosols in the context of Stratospheric Aerosol Intervention. Components of the Earth System (and more in particular, of the atmosphere; i.e., Stratospheric dynamics) are in boxes: for each of them, the main processes that would affect (and be affected by) the injection of $SO_2$ in the stratosphere are listed (red shading), and interactions between components are represented by arrows, with an explanation in grey. "Stratospheric aerosols" and "Stratospheric heating" are in circles to distinguish them from underlying system components, as they can be considered a single component that is affected and affects multiple things in turn.





a limited deployment of SAI might be considered as a short-term mitigation strategy, with more limited consequences on the environment (Tilmes et al. (2020)).

*Code and data availability.* All data used in this work is available from the Earth System Grid (https://esgf-node.llnl.gov/search/cmip6/)

*Author contributions.* DV performed the analyses and wrote the manuscript. DGM and BK helped with analyses and advised DV throughout the writing process. OB, AJ, LT, MM, MJM, PN, UN, RS and ST performed the simulations and offered valuable comments on the manuscript.

*Competing interests.* The authors declare no competing interests.

*Acknowledgements.* Support for DGM was provided by the National Science Foundation through agreement CBET-1818759. Support for
DV was provided by the Atkinson Center for a Sustainable Future at Cornell University. Support for BK was provided in part by the National Sciences Foundation through agreement CBET-1931641, the Indiana University Environmental Resilience Institute, and the Prepared for Environmental Change Grand Challenge initiative. The Pacific Northwest National Laboratory is operated for the U.S. Department of Energy by Battelle Memorial Institute under contract DE-AC05-76RL01830. This work benefited from the French state aid managed by the ANR under the "Investissements d'avenir" programme with the reference ANR-11-IDEX-0004-17-EURE-0006. AJ was supported by
the Met Office Hadley Centre Climate Programme funded by the UK Government Department for Business, Energy and Industrial Strategy (BEIS) and the UK Government Department for Environment, Food and Rural Affairs (Defra). UN has been supported by the Deutsche Forschungsgemeinschaft Research Unit VollImpact (FOR2820). MM,PN and RS acknowledge support from the European Union's Horizon 2020 research and innovation programme under grant agreement No 820829 (CONSTRAIN) and thank the support of the team in charge of the CNRM-CM climate model. MPI-ESM were performed on the computer of Deutsches Klima Rechenzentrum (DKRZ). The CESM
project is supported primarily by the National Science Foundation. The IPSL-CM6 experiments were performed using the HPC resources of TGCC under the allocations 2019-A0060107732 and 2020-A0080107732 (project gencmip6) provided by GENCI (Grand Equipement National de Calcul Intensif). Supercomputing time for CNR-ESM-2 was provided by the Meteo-France/DSI supercomputing center.



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
