# Peer review of "Identifying the sources of uncertainty in climate model simulations of solar radiation modification with the G6sulfur and G6solar Geoengineering Model Intercomparison Project (GeoMIP) simulations"

_Atmospheric Chemistry and Physics, 2021_

## Author Comment (AC1)

Reviewer comments are in bold. Authors' responses are in blue.
Response to reviewer #1 (Peter Irvine)

This article provides a thorough analysis of the similarities and differences between the responses of the GeoMIP G6 sulfur and G6 solar experiments, and a discussion of the uncertainties involved. The article is very strong and would make a valuable contribution to the literature, and I think it is ready for publication after a few minor points are addressed.

We thank the reviewer for their very nice comments! We address all of his points below.

A very minor point but I would suggest reviewing the use of parentheses as there are an awful lot of them. I would suggest reserving parentheses for those points which are truly not relevant to the thrust of the sentence, and restoring some of the details held in many back into the sentences.

Following this and other comments by all the reviewers, we have tried to simplify some of the longer phrases.

The figures look great but one common issue is that they don't make use of greater than and less than arrows (triangles) on the colorbars. If all data falls within the plotted bounds, perhaps this could be stated or if not these colorbars should be changed.

We changed all the colorbar where the issue was correctly raised by the reviewer.

One area I think that the article could elaborate on is the differences in the response to prescribed stratospheric AOD as compared to fully simulated stratospheric aerosols, covering stratospheric heating, chemistry changes, etc.

We have included in Section 2 a discussion of the differences in response for the models with prescribed aerosols.

Specific comments – Note many of these are suggestions to clarify the text and should be taken on or disregarded as the authors' see fit.

L13 – should these sentences be linked by a colon? They seem distinct points to me.

We've separated the two phrases.

L15 – aerosol's?

We've changed the phrase to make sure it's clear.

L19 – should that be a full stop rather than a colon?

Done

L21 – Is there something that could be said for precipitation change?

We mentioned precipitation changes above, giving some numbers: given the complexity of the response, we don't feel like we could say anymore without going into too much detail.

**L38 – missing space, 4x.**

Fixed.

**L60 – by the stratospheric circulation**

Fixed.

**L62 – missing close bracker.**

Added.

**L70 – perhaps add a short phrase linking this list of analysis to the goal of exploring these uncertainties.**

Added.

**L71 – experimental**

Fixed.

**L80 – distributions**

Corrected.

**L81 – put parentheses and cites after "stratospheric processes"**

Done.

**L93 – I think here or elsewhere it is worth reflecting on the view that RCP8.5 is not just a high emissions scenario but an implausibly high emissions scenario, or at least adding a few words of caution around this scenario. This might be raised in the conclusion or introduction instead.**

We had included some comments about this in the conclusions, but we have included some more information on that as suggested.

**L94 – I'd suggest parentheses have been overused in this document, here for example.**

We've removed them here and elsewhere.

**L95 – drop spatially, put the parenthetical statement between commas.**

Done.

**L105 – Perhaps flip the order of this sentence to make it easier to follow: "The teams updated the reduction in solar constant, and the prescribed aerosols … at different intervals, two did so every decade, …."**

We prefer to keep the order of the phrase like this, since then we specify what CESM2 did in the next phrase and it makes more sense.

**L112 – to within 0.2 C of SSP2-4.5 levels.**

Added

**L114 – there are a variety?**

Changed.

**L115 – produce a large spread for the two scenarios.**

Changed.

**Figure 1 – there's a lonely degree symbol, should it just be a K? I'd guess that the ensemble members have been plotted for each model but this isn't stated.**

It should be °C (it's not anomalies). We added a note about the single realizations, thank you for noticing.

**L126 – What about radiative forcing? Surely the response to CO2 forcing and solar forcing are related, even if the % change in insolation and the response per doubling of CO2 are not.**

In a transient climate, the relation might not be obvious (especially considering that there are other evolving forcings, like tropospheric aerosols and other shot-lived GHGs). Some more in-depth analyses could be done with the G1 experiment (described in Kravitz et al., 2021), but it falls outside the scope of this work.

**L130 – Figure 2f? And does it show this? Where is the model spread without normalization? Is this a generalizable result or a chance occurrence due to the make-up of ensemble?**

We fixed the figure label. We have included in the revised version of the supplementary the non-normalized version of Fig. 2f. Due to the modest amount of models, we're not sure we can  say if this is a generalizable result.

**Figure 2d – R-squared is 0.0, is that right?**

Yes.

**L140 – Should that be the other way round, incoming minus outgoing, i.e. positive = more energy input to the earth system?**

Thank you for catching that, fixed.

**L142 – a much more consistent**

Thank you. Fixed.

**L144 – two forcings**

Corrected.

**L145 – rephrase? And drop "indeed"**

Changed.

**L146 – is necessary**

Added

**L146 – drop indeed**

Done

**L148 – more in general? More generally?**

Changed

**L161 – models'**

Fixed.

**L171 – new paragraph?**

Done

**L178 – can't be made.**

Corrected.

**L183 – Are there papers comparing the simulated Pinatubo response and observed Pinatubo response for these models? That would be a valuable point of comparison.**

Sadly, there aren't, although some models will take part in other experiments (like VolMIP and ISA-MIP). These experiment aim to standardize the initial meteorological conditions before the eruption (QBO, winds, ENSO etc.) so that the differences in transport and microphysics can be analyzed. One of these experiments is being analyzed right now, for Pinatubo (https://meetingorganizer.copernicus.org/EGU21/EGU21-13387.html) but results are not final yet (and some models are still missing)

**Figure 4 – the Pinatubo box is clunky and hard to read, is there a better way to present this information? Here and in other relevant figures, it's probably worth mentioning that the yellow and orange lines fully overlap. Panel c – the panel title is too long, how about: "G6 sulfur AOD compared with Pinatubo". In the caption: "the year with a global AOD CLOSEST to that OF Pinatubo"**

Done! Thanks for the suggestions.

**L194 – this sentence is hard to follow, consider revising.**

We rewrote this sentence.

**L200 – this last sentence is also a little muddled, consider revising.**

Rewrote.

**L21 – I think this is reasonable, isn't there a citation to back that up?**

We're not sure what the reviewer is referring to, as the line is probably wrong?

**L214 – this comparison is not fair, the forcing from Pinatubo has had only a year to act, whereas for G6sulfur it has had a few decades. This suggests that the simulated response may be weaker than the observed Pinatubo response.**

Given the feedbacks from other reviewers, we have tried to explain further some of the limitations of comparing with Pinatubo.

**Figure S1 – are there negative AOD values?**

Considering they are derived by considering the AOD in G6 minus that from the background ssp585 experiment, it is possible that some of the initial years produce a local negative result.

**Figure 5 – Is SSP2-4.5 a typo? Should that be SSP5-8.5?**

Yes! Thank you for noticing.

**246 – difference rather than differences. And should that be radiative perturbation from CO2.**

Fixed.

**L250 – stratospheric heating repeated?**

Yes. We removed the repetition.

**L251 – is this inability to restore the ocean state of the same character as the other things listed? It seems to be a consequence rather than a driver of differences in response. Though perhaps I've misunderstood the point.**

It would depend on what's driving the particular feature of the oceanic circulation (of the AMOC, in particular). And even more important, what is the driver in the models (which could be different) compared to the real world.

**L255 – Is Macmartin the right reference to introduce this analysis? Won't the main point of Macmartin still hold? I.e., that the uncertain degree of warming under RCP8.5, that is absent or reduced in solar geo scenarios, is itself associated with regional climate changes and hence drives model spread (and uncertainty). The Macmartin argument applies to the spread in RCP8.5 projections as compared to those of G6.**

In a way it still holds, as we show in Fig. 6, but much less for G6Sulfur than for G6Solar. And, MacMartin et al. (2015) analyses used G1, which is solar dimming over 4xCO2, so it was a much simpler experiment in many ways.

**L260 – I'd suggest not switching the order of comparison here, i.e., the spread is larger in G6 sulfur.**

Fixed.

**L261 – I think it is a mischaracterization to say that G6sulfur is unable to cool down the Norther high latitudes. It looks to me as if ~2/3 of the warming difference has been offset. This should be rephrased to make clear that it is at least partially effective.**

We added the word "completely" to better clarify the point.

**L263 – Is this definitely due to heating rather than some other factor?**

It is one of the possible causes (and in CESM we have verified in other works that this is the case). But we've toned down the phrase.

**L265 – although, if I recall correctly it's of smaller magnitude than would be expected from just the difference in radiative forcing due to countervailing circulation changes, right?**

Yes. We have fixed the phrase.

**Figure 6 – Why do the colobars not have "greater than" extensions? The lack of such extensions suggests that all points in the arctic in 6c, for example, see less than 1.1C of warming. Is that correct? Should note the change in colorbar range between a and c and e.**

See response above. We changed all the colorbars.

**Figure 7 – no colorbar label.**

Added.

**L305 – point to?**

Corrected

**L333 – Should this be split into 2 sentences?**

Done!

**L345 – tends.**

Fixed.

**L346 – will occur over the tropics, rather than observed.**

Changed.

**L358 – this last sentence is very long, consider splitting. Should that be "direct effects of CO2", as in the CO2 physiological effect or is this also CO2 radiative effects too? Changes in dynamical changes?**

We have changed the phrase to make it clearer.

**L360 – Could this effect be quantified, i.e., what's the land-mean precipitation error for those two projections?**

We have included some discussion and a supplementary figure of the land-mean precipitation.

**Table 1 – CMIP6 doi? Is that a typo for MPI models, i.e., should it read: "r1,r2,r3" and not "r1,r2,r"?**

The DOI (sorry, we changed that to DOI from doi) is the one for the various datasets uploaded to CMIP6. We fixed the problem.

**Table 2 – should "last decade (2081-2100)" be "last 2 decades (2081-2100)"?**

Fixed.

**Figure 10 – The side-panels don't line up with the maps and it's hard to read given the number of colourful lines, I'd suggest pulling them out to a separate figure as little is gained from combining them with the maps. I also would only show one experiment for absolute plots.**

We have separated the two figures.

**L370 – "reduction in global-mean precipitation" should be elaborated to make clear what that reduction is relative to, as there is also a reduction seen in SSP2-4.5 relative to SSP5-8.5.**

Done!

**L372 – This sentence has a strange structure, consider revising.**

We've tried to clarify the phrase.

**L385 – no need to specify air, although it could be specified in air temperature.**

Removed.

**L386 – this second half could be rephrased.**

Done!

**L415 – (and more) at wrong point.**

Changed.

**L422 – rephrase - not even the most ideal.**

Done

**L425 – restoring or maintaining the ITCZ location?**

Changed.

**Figure 11 – stratospheric aerosol intervention – first time used. "more particularly". Just a suggestion, but this might be easier to follow if the boxes were ordered from top to bottom: stratospheric heating and aerosols top (the inputs) à stratospheric chem, dynamics and aerosol chem / microphysics à Radiative forcing + surface climate (the outputs).**

Thanks for the suggestions. We have made the changes in the revised figure.

---

## Author Comment (AC2)

**Reviewer comments are in bold**. Authors' responses are in blue.
**Response to reviewer #2**

**This paper reported one set of the experiments in the GeoMIP – G6solar and G6sulfur using 6 climate model output. Although models successfully reach temperature target (from the level of SSP5-8.5 to the level of SSP2-4.5), there are different climate responses between G6solar and G6sulfur, and large inter-model spreads in many climate aspects. The manuscript raises lots of questions on model uncertainties and following-up impact studies. In general, this study is strong and important as it is the first thorough report on G6solar and G6sulfur from 6 models, and it indicates many future research directions for future. But the manuscript needs some improvements in the writing style before publishing. The author tends to use long and obscure sentences when describing the figures and explaining the underline hypothesizes.**

We thank the reviewer for their encouraging comments, and for their help in improving the clarity of our manuscript, which is really appreciated. We respond to all their points below.

**Specific comments:**
**Line 7-9: it is better to change to something like "We find that, over the two decades of the century, there are considerable inter-model spreads in the needed injection amounts of sulfate, in the latitudinal distribution of the aerosol cloud, and in the stratospheric temperature changes resulting from the extra aerosol layer."**

Thank you for the suggestion. We have changed the phrase accordingly.

**Line 12-13: Are those values averaged differences between SSP5-8.5 and SSP2-4.5 over 2081-2100? If so, it is better to make it easy to read. Something like "SSP5-8.5 minus SSP2-4.5 averaged over 2081-2100"**

They are. We have changed the phrase.

**Line 13: please change ": the differences in the simulated aerosol spread then change some of the underlying uncertainty, for example in terms of" to ". With aerosol injection, the differences of aerosol spread further change some of the underling uncertainties, such as"**

Changed.

**Line 16: please change "a larger inter-model spread in the regional response in the surface temperatures" to "a larger uncertainty in the regional surface temperature response among models"**

Done. Thanks for the suggestion.

**Line 44: please clarify "with no baseline simulation to analyse the response against (as in the case of G4)".**

*We understand this was unclear. We changed that to specify we meant that "In the case of the G4 experiment, furthermore, there was no sustained future scenario with similar global surface temperatures achieved without geoengineering, but with less $CO_2$, to compare the results against"*

**Line 54: "Both reductions" of what? The temperature reductions are checked every decade, and then the sulfate injection amount and solar radiation reduction are adjusted.**

*We added the words "of incoming surface insolation" to clarify.*

**Line 55: please reorganize this sentence "There are multiple uncertainties…intercomparison"**

*Done!*

**Line 62: add ")" after "Visioni et al. (2017b)"**

*Added.*

**Line 64: add "SO2" after "Tg"**

*Done.*

**Table 1: The first column – model names, are hard to read. Maybe add one extra space among models?**

*We have rewritten the models' name in bold for clarity.*

**Line 99: CESM2 also injects SO2 between 10N-10S? Or following the feedback algorithm and injecting SO2 from other latitudes?**

*As a mistake, CESM2 only injected at 0°N.*

**Line 100: how could CNRM-ESM2-1 use SO2 distribution file from G4SSA for G6 experiment?**

*They used the aerosol distribution, not SO2. We have changed the text for clarity.*

**Line 102: change ";" to "and"**

*Fixed.*

**Figure 1: Please keep all sub title styles consistent.**

*Done!*

**Line n 116-120: please reorganize this sentence. It is too long.**

*We have split the sentence up.*

**Table 2: average of which period? In the title, it is said "the last decade of the experiment", and also said "2081-2100". Please use "SSP58.5 minus SSP24.5" instead of "SSP58.5-SSP24.5". And please use minus instead "-" in the whole manuscript.**

We have specified it's the last *two* decades, and tried to change the text according to the reviewer's suggestion when possible (except in the table, for reasons of space).

**Figure 2: R2 in d is zero? What is m?**

Yes, it's 0. We have updated the caption for clarity (m is the slope of the linear fit).

**Line 131-132: how are those numbers calculated? What does this sentence mean?**

Following the request from reviewer 1, we have included in the supplementary the non-normalized version. We have updated the text for clarity.

**Line 144: please indicate which panels a and b are.**

Done!

**Line 145: why the comparison only helps the first 30 years?**

Because some models show very little changes in TOARF in that period (for instance, CESM2) and that explains why the amount of intervention is very small, or constant, in there.

**Line 148: please clarify the sentence. As far as I understand, the first part of the sentence means that the small differences of global mean temperatures anomaly between SSP5-8.5 and SSP2-4.5 among models tend to magnify the inter-model differences of intervention applied. But what does "resulting in larger differences in the first years" mean?**

We have tried to clarify. We mean that, with small forcings, the estimates made by the single modeling teams would have magnified some of the differences.

**Line 148, 150: please change "first decades" and "first years" to "first couple decades" and "first several years". Or indicate exact numbers.**

Done.

**Line 145-153: I still don't understand the different mechanisms behind the two periods (first three decades and the rest)**

We hope the clarifications in the text have helped. For instance, if 0.3 W/m$^2$ in TOARF have to be balanced in the first decades, and the sensitivity of the estimate is around 0.1 W/m$^2$, the error can be large. Towards the end, if the imbalance is 2.0, the same sensitivity influences less the results.

**Line 161: "different models' behavior"**

Fixed.

**Line 164: "reasons for a different aerosol distribution with similar injection locations and height of SO2 are"**

Changed.

**Figure 4: CNRM-ESM2-1 and MPI-ESM are both prescribed SO2 distribution files. How could MPI-ESM have SO2 injection amount, but CNRM-ESM2-1 not in b)? The box is hard to read. It is better to draw it a table. If all three panels use the same color code, then only one legend is needed.**

MPI derived their aerosol distribution from a similar version of the model without interactive ocean or land (to save computational time): it is therefore consistent to determine the $SO_2$ amount from the scaling of the required AOD: CNRM uses another model's distribution, therefore it wouldn't be. We fixed the figure as suggested.

**Line 183: please clarify this sentence. "Model spread, … observations)."**

Uncertainty is related to the distance between simulations and real world (also known as accuracy). Models' spread is not, therefore, the same as uncertainty. We have clarified the phrase changing it to *"Model spread on a particular result is not, of course, the same as uncertainty: models may agree despite a lack of observational support, resulting in a narrow spread that might be inaccurate, or the spread might be large because some model results are simply inconsistent with available observations"*.

**Line 209-210: are those values (e.g. CESM is 6.2 at 2058) in models accumulated SO2 or SO2 per year? Based on Fig. 4b, they are SO2 Tg/yr? If so how this annual injection amount compare to one time injection from Pinatubo? Before the AOD reaching the level of that in Pinatubo, there have been decades of injection already.**

They are annual. It is, of course, a rough comparison, but given a 1 year lifetime for stratospheric aerosols, they can be somewhat compared.

**Line 231: UKESM1-0-LL is showing much weaker stratospheric heating, but from Figure 5, it seems that CNRM is the least? Also IPSL shows similar stratospheric heating as in MPI.**

We corrected the phrase.

**Line 251: "Banerjee et al. (2020))"**

Fixed.

**Line 259-261: please reorganize this sentence "not only are …. Removed"**

Done

**Line 261: "For G6sulfur, there is a general model agreement in the …"**

Changed.

**Line 264: "as evidenced by the absence of high-latitude warming with the same magnitude in the G6solar simulations"**

Done.

**Line 275: please reorganize the first sentence.**

Changed.

**Line 277: why saying "that is balanced in the multi-model average by a stronger warming modeled by MPI-ESM"? all models are showing the warming over Northern Eurasia with different magnitudes. There is nothing to be balanced.**

The reviewer is right. We removed the phrase.

**Line 298-299: Both versions of MIP show the warming in North America and West Antarctica, just one is much stronger than the other.**

Fixed.

**Line 299-300: How could the different responses in HR and LR indicate that the deep ocean circulation is causing regional differences in temperature? What is the difference between HR and LR?**

The difference is just in the horizontal resolution, so this is just our assumption. We hope to investigate this further in the future!

**Line 336 to 344: CESM seems showing the least difference between G6sulfur and G6solar. Actually they are almost the same amount of reduction.**

Changes in precipitation are much larger East of Australia, for CESM2, however. We added a note anyway.

**Line 345: please reorganize this sentence to something like "Lastly, models agree on regional precipitation changes more in G6solar than in G6sulfur"**

Done!

**Figure 10: legends cover part of the plots.**

To do

**Line 351-352: HR shows stronger reduction and increasing of precipitation in the tropics relative to LR, not just stronger reduction.**

Fixed.

**Line 354: Again, why are circulations different in HR and LR?**

We added a note specifying that the horizontal resolution is different (thus we can assume the circulation behaves differently).

**Line 358-363: please reorganize this sentence.**

Done!

**Figure 11: Hard to read. Some suggestions: 1. Order the main categories with altitude: Surface at the bottom, Stratosphere on the top, Radiative forcing and Aerosol chemistry/microphysics are through the whole troposphere and stratosphere. 2. "Stratospheric aerosols" and "Stratospheric Heating" should be in the same category. Stratospheric aerosol is what we injected in the model, and all others (including stratospheric heating, dynamics and chemistry…) are responses to the extra aerosols**

Thanks for the suggestions. We have revised the figure accordingly.

---

## Author Comment (AC3)

**Reviewer comments are in bold**. Authors' responses are in blue.
**Response to reviewer #3**

**This manuscript investigates the difference between simulations where solar radiation management is implemented as a reduction of incoming solar radiation and as an injection of stratospheric aerosols. The authors find that the two methods lead to a different response in the surface climate, and therefore are not equivalent in simulating geoengineering (as suggested in a few previous papers).**
**I have found this paper interesting, clear, and well written. I only have very minor comments.**

Thank you for the kind words and for the suggestions! We have responded to each point below.

**L45:  I supposed with "baseline simulation" the author means a simulation of a desirable climate reached via emission mitigation. My first understanding, though, was that a baseline simulation was one without geoengineering, which would not be correct because I believe G4 also required a simulation without geoengineering.**

Yes, we have corrected this to be more clear.

**L49: The discriminant is not the presence of microphysics, but rather the presence of a sulfur cycle, or a way to represent the formation of new particles from sulfur dioxide. A microphysics representation allows for the evolution of particle size, which is one additional process beyond the formation of new particles.**

Corrected.

**L130: Shouldn't the reference be to Fig. 2f?**

Yes. Fixed.

**L132: what does it mean that the solar reduction and the AOD seem unrelated? In Fig. 2e you show a fit with R^2 of 0.72. The word "unrelated" should be replaced with a quantitative metric**

We have changed this as suggested.

**L142: eliminate "much"**

Done.

**L147: can you explain better how this is a feature of the current feedback controller? I understand that you refer to Tilmes et al. (2018) but I think it would be more complete if there was a not too specific explanation here without having to look for another paper**

Of course. Done.

**L152: I don't understand the connection between the sentences before and the conclusion that they explain why the AOD in 2050 is the same among all models.**

We agree that it didn't sound very clear, and have removed the phrase.

**L160: Does Solar produce any difference in the stratosphere? I imagine that reducing the solar flux would impact UV absorption by ozone (hence the temperature profile) as well as ozone concentrations**

It produces minimal changes, as we have shown for CESM1 in Visioni et al. (2021). We plan to analyze some of the stratospheric changes in these models in the future.

**L189: I'm really not sure how meaningful is the comparison between Pinatubo and geoengineering simulations, and I would not include it at all. As you write here, this is a sustained injection, and you already mentioned several times that Visioni et al. showed the importance of the injection seasonality on the transport. Jones et al. (2016, doi:10.1002/2016JD025001) showed the dependence of the Pinatubo dispersal from the initial conditions. What I find incorrect is that this is the only "real" point of comparison. The only real point of comparison is a simulation of Pinatubo (which all models could do - and probably have done already) vs observations of Pinatubo. That would be informative with respect to the ability of each models to simulate the transport of stratospheric aerosols.**

Yes, the initial conditions are fundamental. This is also why looking at Pinatubo in i.e. the Historical simulations would also be pretty imperfect, as they did not check for the proper meteorological conditions in those simulations, for Pinatubo. There is an experimental protocol (ISA-MIP that is in the process of doing some more robust comparison, but the results are yet to come (see https://meetingorganizer.copernicus.org/EGU21/EGU21-13387.html). Our point here was just to have a pretty rough comparison between models considering the same achieved global AOD as Pinatubo. We've toned down parts of the discussion to be more clear.

**L200: Is this true also if you calculate the spread as % of multimodel mean? I am not sure that the better agreement isn't simply a result of the AOD values being smaller in 4c than 4a**

Yes, that remains true anyway.

**L210: It would be interesting to show the optical properties (the extinction efficiency) used by each model and/or the simulated particle size distribution**

We agree. Sadly, the variables needed for that were not made available by the modeling teams (as the microphysical properties of the aerosols were not the main focus of CMIP6, so they were not between the necessary variables to upload).

**L214: Canty et al. (2013, https://doi.org/10.5194/acp-13-3997-2013) finds a decrease of 0.14°C globally.**

Thank you for the valuable reference! We have updated the discussion accordingly.

**Fig. 7: I am a bit confused by the statistics. The ensembles contain two, three at the most, ensemble members. Did you perform a T test with only 2 realizations per experiment?**

We considered a set of 20 years X the number of ensemble members for the test. We fixed the caption for clarity

References

Visioni, D., MacMartin, D. G., and Kravitz, B.: Is Turning Down the Sun a Good Proxy for Stratospheric Sulfate Geoengineering?, Journal of Geophysical Research: Atmospheres, n/a, e2020JD033 952, https://doi.org/https://doi.org/10.1029/2020JD033952, https://agupubs. onlinelibrary.wiley.com/doi/abs/10.1029/2020JD033952, e2020JD033952 2020JD033952, 2021.